# Association between *GnRH* Receptor Polymorphisms and Luteinizing Hormone Levels for Low Ovarian Reserve Infertile Women

**DOI:** 10.3390/ijerph18137006

**Published:** 2021-06-30

**Authors:** Shun-Long Weng, Shu-Ling Tzeng, Chun-I Lee, Chung-Hsien Liu, Chun-Chia Huang, Shun-Fa Yang, Maw-Sheng Lee, Tsung-Hsien Lee

**Affiliations:** 1Department of Obstetrics and Gynecology, Hsinchu MacKay Memorial Hospital, Hsinchu 30071, Taiwan; 4467@mmh.org.tw; 2Department of Medicine, MacKay Medical College, New Taipei 25245, Taiwan; 3Mackay Junior College of Medicine, Nursing and Management College, Taipei 11260, Taiwan; 4Institute of Medicine, Chung Shan Medical University, Taichung 40203, Taiwan; cherie@csmu.edu.tw (S.-L.T.); ysf@csmu.edu.tw (S.-F.Y.); msleephd@gmail.com (M.-S.L.); 5Department of Medical Research, Chung Shan Medical University Hospital, Taichung 40203, Taiwan; 6Department of Obstetrics and Gynecology, Chung Shan Medical University Hospital, Taichung 40203, Taiwan; adoctor0402@gmail.com (C.-I.L.); chliu@csmu.edu.tw (C.-H.L.); 7Department of Medicine, School of Medicine, Chung Shan Medical University, Taichung 40203, Taiwan; 8Division of Infertility Clinic, Lee Women’s Hospital, Taichung 40602, Taiwan; agarhuang@gmail.com

**Keywords:** *GnRH* receptor, anti-Müllerian hormone, assisted reproduction technology, single nucleotide polymorphism, poor responders, POSEIDON criteria, *GnRH* antagonist, *GnRH* agonist

## Abstract

The choice of ovarian stimulation protocols in assisted reproduction technology (ART) cycles for low ovarian reserve patients is challenging. Our previous report indicated that the gonadotrophin-releasing (GnRH) agonist (GnRHa) protocol is better than the *GnRH* antagonist (GnRHant) protocol for young age poor responders. Here, we recruited 269 patients with anti-Müllerian hormone (*AMH*) < 1.2 ng/mL undergoing their first ART cycles for this nested case-control study. We investigated the genetic variants of the relevant genes, including follicular stimulating hormone receptor (*FSHR*; rs6166), *AMH* (rs10407022), *GnRH* (rs6185), and *GnRH* receptor (*GnRHR*; rs3756159) in patients <35 years (*n* = 86) and patients ≥35 years of age (*n* = 183). Only the genotype of *GnRHR* (rs3756159) is distributed differently in young (CC 39.5%, CT/TT 60.5%) versus advanced (CC 24.0%, CT/TT 76.0%) age groups (recessive model, *p* = 0.0091). Furthermore, the baseline luteinizing hormone (LH) levels (3.60 (2.45 to 5.40) vs. 4.40 (2.91 to 6.48)) are different between CC and CT/TT genotype of *GnRHR* (rs3756159). In conclusion, the genetic variants of *GnRHR* (rs3756159) could modulate the release of LH in the pituitary gland and might then affect the outcome of ovarian stimulation by GnRHant or GnRHa protocols for patients with low *AMH* levels.

## 1. Introduction

In assisted reproduction technology (ART) for infertile couples, management for patients with an inadequate ovarian response is challenging [1,2,3,4]. Although investigators propose two international definitions, the Bologna [5] and POSEIDON [6] criteria, for poor responders, the choice of ovarian stimulation protocols for these patients is still controversial [7,8]. Most reports recommended the protocol for the use of gonadotropin-releasing hormone (*GnRH*) antagonist (GnRHant) or the mild stimulation protocol (MSP) for the poor responders, instead of the long protocol for using *GnRH* agonist (GnRHa) [9]. The primary reason for this recommendation is the cost-effective consideration, which is adjusted by the cost per oocyte retrieved and the number of exogenous gonadotropin injections. However, because it remains controversial, the GnRHa protocol is less recommended for poor responders.

The number of retrieved oocytes and patients’ age are the key predictors for a successful pregnancy and live birth in ART cycles [1]. The aneuploidy rates of oocytes/embryos are intimately correlated with maternal age. This means that oocyte quantity and quality (euploidy) are the critical factors for successful ART cycles. Our previous study, however, indicated that for young patients (<35 years of age) with low anti-Mullerian hormone (*AMH*) levels, the GnRHa protocol is better than the GnRHant protocol in terms of embryos available for transfer and pregnancy rate [10]. A randomized clinical trial also reported that the GnRHant protocol is correlated with a higher cancellation rate than that of the GnRHa protocol [11]. Those results suggested that the follicular and the subsequent embryo development after controlled ovarian stimulation might differ between GnRHant and GnRHa protocols for poor responders.

The response (number of retrieved oocytes) to ART protocols could be predicted by ovarian reserve markers, such as *AMH* or antral follicle count (AFC) [12,13]. However, some researchers observed unexpected poor ovarian responders after controlled ovarian stimulation in ART cycles, such as POSEIDON group 1 and group 2 patients. The single nucleotide polymorphism (SNP) in the hormones and hormone receptors related to follicular growth and development might cause such inadequate ovarian response [14]. For example, SNPs of *AMH* [14,15], follicular stimulating hormone (FSH) [16,17], and FSH receptor (*FSHR*) [16,17,18,19,20,21] have been surveyed to explain the ovarian response subsequent to controlled ovarian stimulation in ART cycles [14,17].

We previously reported a better pregnancy outcome by the GnRHa than the GnRHant protocol in POSEIDON group 3 patients [10]. Therefore, we raised the hypothesis that, in addition to the SNPs of *AMH* (rs10407022) and *FSHR* (rs6166), the SNPs of *GnRH* (rs6185) [22] or *GnRH* receptor (*GnRHR*; rs3756159) [23], may account for the difference of embryo development and pregnancy outcome for patients with low *AMH* levels (POSEIDON group 3 and group 4). The present study results revealed that the distribution of SNP of *GnRHR* (rs3756159) varied between POSEIDON group 3 and group 4 patients, and those SNPs are associated with varied baseline LH levels in ART cycles.

## 2. Materials and Methods

### 2.1. Study Design and Patient Selection

The infertile couples who underwent their first ART treatment cycle from January 2014 to December 2015 were recruited for this prospective nested case-control study. The inclusion criteria were as follows: (1) women age <45 years old; (2) serum *AMH* <1.2 ng/mL before ART treatment; and (3) no histories of ovarian surgery or pelvic radiation treatment. We drew a venous blood sample for DNA extraction and subsequent analysis for the chosen SNPs. The Institutional Review Board of Chung Shan Medical University Hospital approved the study protocol (CS13194 and CS2-14033). A written informed consent was obtained from each participant. All the recruited women for this analysis were Han Chinese people. Clinical trial register number: ISRCTN12768989.

We attempted to study the SNPs of related hormone molecules for ovarian responses in the GnRHa and GnRHant protocols. The methods for chosen SNPs were in line with our previous report [24], which was based on the searches in dbSNP (http://www.ncbi.nlm.nih.gov/snp, accessed 3 November 2013) and the international HapMap project (http://hapmap.ncbi.nlm.nih.gov, accessed 3 November 2013). Consequently, we surveyed the SNPs of *AMH* 146 T > G (rs10407022), *FSHR* Asn680Ser (rs6166), *GnRH* (rs6185), and *GnRHR* (rs3756159). The *AMH* 146 T > G is at the coding region of *AMH* and causes amnio acid substitution. *FSHR* Asn680Ser (rs6166) is the most common reported SNP that related to ovarian responses in ART cycles [16,17,18,19,20,21]. *GnRH* (rs6185) and *GnRHR* (rs3756159) are chosen to discriminate the different activities of GnRHa and GnRHanta in ART treatment. *GnRH* (rs6185) is located at the coding region and results in amnio acid substitution. *GnRHR* (rs3756159) is located at position 68305073 on chromosome 4 in the 5’ untranslated region of the *GnRHR* gene.

### 2.2. ART Treatment Protocol and Hormone Analysis

During the study period from January 2014 to December 2015, we recruited those patients undergoing the same GnRHa stimulation protocol to avoid bias in the association between the chosen SNPs and ART outcomes. The ovarian stimulation procedure is the same as previously described [24]. The GnRHa protocol comprises daily injections of 0.5 mg of leuprolide acetate (Lupron; Takeda Pharmaceutics, Konstanz, Germany) from the mid-luteal phase (cycle day 21) of the previous cycle. After that, recombinant FSH (Gonal-F, Merck-Serono, Darmstadt, Germany) or highly purified FSH (Menopur; Ferring Pharmaceuticals) was administered daily for follicular growth. We used 10,000 IU human chorionic gonadotropin (Profasi, Serono, Norwell, MA, USA) to trigger final oocyte maturation, and ovum pick-up was performed 36 to 38 h later.

The baseline *AMH*, FSH, luteinizing hormone (LH), and estradiol (E2) levels were measured on day 2 to 3 of the menstruation cycles before the controlled ovarian stimulation. On day 2 to 3 of the hyper-stimulation cycle prior to gonadotropin injection, FSH, LH, and E2 levels were assessed again. On the day of hCG trigger, measurement of E2, LH, and progesterone (P4) levels was performed. The *AMH*/MIS ELISA kit (Immunotech/Beckman Coulter Inc., Marseille Cedex, France) was used in duplicate to assess serum *AMH* levels. A specific immunometric assay kit (Access; Beckman Coulter Inc., Fullerton, CA, USA) was utilized to measure serum FSH, LH, E2, and P4 levels. The sensitivity, intra-assay coefficient of variation (CV), and interassay CV for the FSH measurement were 0.2 mIU/mL, 4.3%, and 5.6%, respectively. The sensitivity, intra-assay CV, and interassay CV for the LH evaluation were 0.2 mIU/mL, 5.4%, and 6.4%, respectively.

### 2.3. DNA Extraction and Determination of Genotypes

We obtained genomic DNA with a QIAamp DNA blood mini kit (Qiagen, Valencia, CA, USA) from EDTA anti-coagulated venous blood. Genomic DNA extraction was performed following the manufacturer’s instructions as in a previous report [25]. We used Tris-EDTA (TE) buffer to disperse the extracted DNA and then measured the optical density at 260 nm to determine DNA quantity. The final solution was stored at −20 °C and used as polymerase chain reaction (PCR) templates. Genotyping of the four studied SNPs was assessed with the ABI StepOne™ Real-Time PCR System (Applied Biosystems, Foster City, CA, USA), and allele discrimination was analyzed using SDS version 3.0 software (Applied Biosystems, Foster City, CA, USA) and the TaqMan assay (Applied Biosystems, Foster City, CA, USA) [26]. The primers used for each genotype are listed in Table 1.

### 2.4. Statistical Analysis

The data supplemental to this prospective study is listed in the Appendix A: SNP_low_*AMH*_single.txt A chi-square test was performed to determine the Hardy–Weinberg equilibrium, including *AMH* (rs10407022), *FSHR* (rs6166), *GnRH* (rs6185), and *GnRHR* (rs3756159). Then, a chi-square test examined the associations between POSEIDON group 3/4 and tested SNPs under the genotypic (AA versus Aa versus aa) and recessive (AA versus Aa/aa) models.

The Kolmogorov–Smirnov test was used for the demographic characteristics and other clinical parameters about ovarian responses to determine the distribution of those variables. After that, the continuous variables are presented as medians (interquartile range (IQR, 25th–75th percentile)), whereas categorical variables are shown as numbers and percentages. We used the Mann–Whitney U test (for continuous variables) or chi-square test (for categorical items) to compare the differences between groups with genetic variants under the recessive model (AA versus Aa + aa). All data were analyzed using the IBM SPSS Statistics for Windows, Version 22.0 (IBM Corp., Armonk, NY, USA). *p*-values < 0.05 were considered statistically significant.

## 3. Results

A total of 269 patients with low *AMH* levels undergoing their first IVF/ICSI cycles were recruited for this nested case-control study in a prospective cohort. The 269 patients were divided into young (<35 years of age, POSEIDON group 3, *n* = 86) and advanced (≥35 years of age, POSEIDON group 4, *n* = 183) age groups.

The representative results of real-time PCR and TagMan assay for each SNP genotype are shown in Figure 1 (for *AMH* 146 T > G (rs10407022)), Figure 2 (for *FSHR* A2039G (rs6166)), Figure 3 (for *GnRH*-1 (rs61850)), and Figure 4 (for *GnRHR*-1 (rs3756159)).

### 3.1. The Distribution of AMH, FSHR, GnRH, and GnRHR

The SNPs in *AMH* 146 T > G (rs10407022; Table 2), *FSHR* A2039G (rs6166; Table 3), and *GnRH*-1 (rs6185; Table 4), were not correlated with patients with diminished ovarian reserve phenotype. Only SNP in *GnRHR*-1 (rs3756159) was distributed differently in young (CC 39.5%, CT/TT 52 (60.5%)) versus advanced (CC 24.0%, CT/TT 139 (76.0%)) age groups (recessive model, *p* = 0.0091; Table 5).

### 3.2. The Clinical Characteristics of Patients with Varied Genotypes of GnRHR SNP (rs3756159)

Table 6 demonstrated the patients’ clinical parameters with the varied genotypes of *GnRHR* SNP (rs3756159). Among the clinical parameters related to ART cycles, only the baseline LH levels (3.60 (2.45 to 5.40) vs. 4.40 (2.91 to 6.48)) are different between the CC and CT/TT genotypes of *GnRHR* SNP (rs3756159).

Figure 5 revealed the baseline LH levels in POSEIDON group 3 and 4 patients divided by *GnRHR* SNP (rs3756159) genotypes (CC vs. CT/TT). For POSEIDON group 3 patients, the baseline LH levels are 3.40 (2.40 to 5.30) in the CC vs. 4.40 (2.85 to 6.46) in the CT/TT genotypes (*p* = 0.095 by Mann–Whitney U test). Furthermore, the baseline LH levels are 3.65 (2.70 to 5.90) in the CC vs. 4.50 (2.93 to 6.48) in the CT/TT genotypes (*p*= 0.152 by Mann–Whitney U test) in POSEIDON group 4 patients.

## 4. Discussion

In the present study, we found that the young patients with low *AMH* levels (POSEIDON group 3) are associated with a higher frequency of wild-type CC of *GnRHR* (rs3756159). Interestingly, we also noted a lower baseline of serum LH in patients with CC *GnRHR* (rs3756159). The results indicated that *GnRHR* SNP rs3756159 distributed significantly differently between POSEIDON group 3 and group 4 patients and might modulate the ovarian responses using GnRHa or GnRHanta protocols.

The primary physiological function of *GnRH*-*GnRHR* signaling is to release FSH and LH from the pituitary gland. The high baseline serum LH in patients with the CT/TT *GnRHR* (rs3756159) genotype indicated that the *GnRHR* (rs3756159) might modulate the function of *GnRH* on the target organs or tissues. The *GnRHR* (rs3756159) is located at the 5′ upstream untranslated region of the *GnRHR* gene. How the genetic variants affect the protein structure or function of *GnRHR* deserves further investigation. Interestingly, our previous report also showed higher LH levels in POSEIDON group 4 than those in POSEIDON group 3 for patients with GnRHa or GnRHant protocols [10]. Both reports indicated that the high LH levels in POSEIDON group 4 patients might correlate with the genetic variants of *GnRHR* (rs3756159). Furthermore, such different LH levels might affect follicular and embryo development [27,28,29].

The use of LH-containing agents, such as human menopausal gonadotropins (HMG), could improve the live birth rate for POSEIDON group 3 and 4 patients [27]. Nonetheless, the supplement of recombinant LH for IVF patients is beneficial for women 36–39 years of age but not young (<35 years) normal responders in a recent systemic review [28]. Furthermore, the GnRHa protocol is associated with a deeper suppression of LH, the supplementary of recombinant LH is no benefit for young patients in that meta-analysis [28]. It seems that the benefit of high serum LH levels or supplementing recombinant LH is only exhibited among women >35 years of age. By contrast, for women <35 years, the GnRHant protocol is associated with a higher rate of cancellation in our previous report for poor responders [10] and a large retrospective analysis by Grow et. al. in 2014 for women with a good prognosis [29].

The *GnRHR* (rs3756159) features a modulation effect of LH release from the pituitary gland in the present study. Although the *GnRHR* (rs3756159) CT/TT genotype is associated with a higher baseline serum LH levels and more common in POSEIDON group 4 patients, the benefit of high LH levels is demonstrated for women 36–39 years of age, in other words, the POSEIDON group 4 patients. These might partially explain why the GnRHa protocol’s performance was better than the GnRHant protocol for POSEIDON group 3 patients, but the efficacy of these two protocols is almost equal for POSEIDON group 4 patients in our previous report [10].

The limitation of the present study includes a relatively small sample size, and we did not recruit patients with GnRHant protocol. The baseline LH levels are higher in patients with *GnRHR* (rs3756159) CT/TT genotype than those with CC genotype either POSEIDON group 3 or group 4 patients, but the difference did not reach statistical significance (Figure 1). Nonetheless, the effect of *GnRHR* genotype on baseline LH levels is exhibited before the commence of controlled ovarian stimulation, no matter which stimulation protocol would be used.

The *FSHR* SNPs, including A2039G (rs6166), are the most common studied genetic variants related to ovarian response in ART treatment [16,17,18,19,20,21]. A recent meta-analysis indicated that FRHR rs6166 is associated with premature ovarian insufficiency (POI) in an Asian population [20]. In the present study, although most of the population is younger than the age of 40 (for the definition of POI), the reference group consists of poor responders < 35 years of age instead of normal responders. Furthermore, the ovarian dysfunction in the patients we recruited for this prospective study is not as severe as those in POI patients. That may explain why *FSHR* rs6166 is not different between POSEIDON group 3 and group 4 patients in the present study.

## 5. Conclusions

The genetic variants at *GnRHR* (rs3756159) distributed differently between POSEIDON group 3 and group 4 patients. The CC phenotype is more common in POSEIDON group 3 patients than in POSEIDON group 4 patients. Furthermore, the CC phenotype is associated with lower LH levels compared with the CT/TT phenotype. The *GnRHR* (rs3756159) may modulate the ovarian responses using GnRHa or GnRHant protocols. The effect of *GnRHR* (rs3756159) on ovarian responses and embryo development deserves further investigation.

## Figures and Tables

**Figure 1 ijerph-18-07006-f001:**
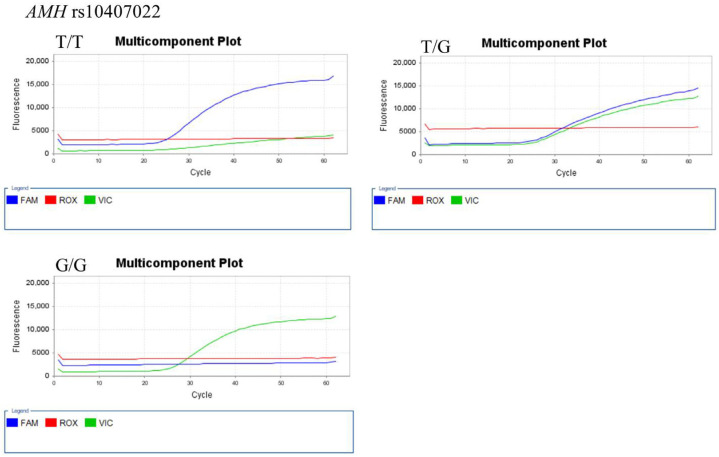
Representative TaqMan assay for *AMH* rs10407022 genotyping. The FAM (blue) and VIC (green) fluorescence probes detect T and G alleles, respectively. The ROX (red) fluorescence probes are used for calibration.

**Figure 2 ijerph-18-07006-f002:**
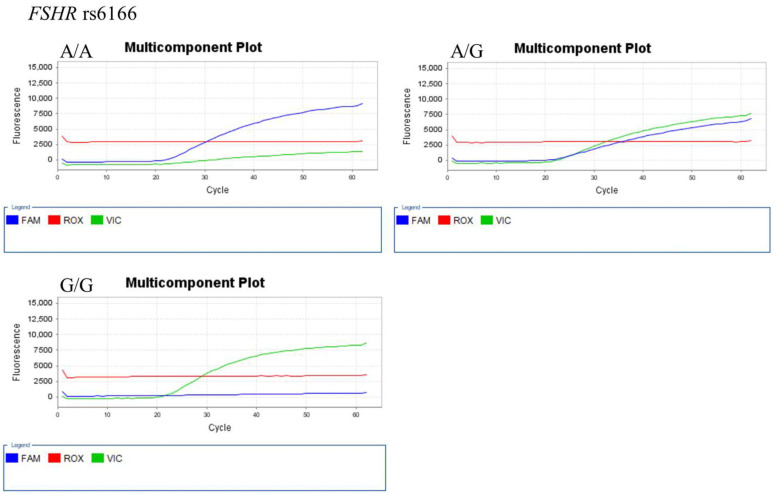
Representative TaqMan assay for *FSHR* (rs6166) genotyping. The FAM (blue) and VIC (green) fluorescence probes detect A and G alleles, respectively. The ROX (red) fluorescence probes is used for calibration.

**Figure 3 ijerph-18-07006-f003:**
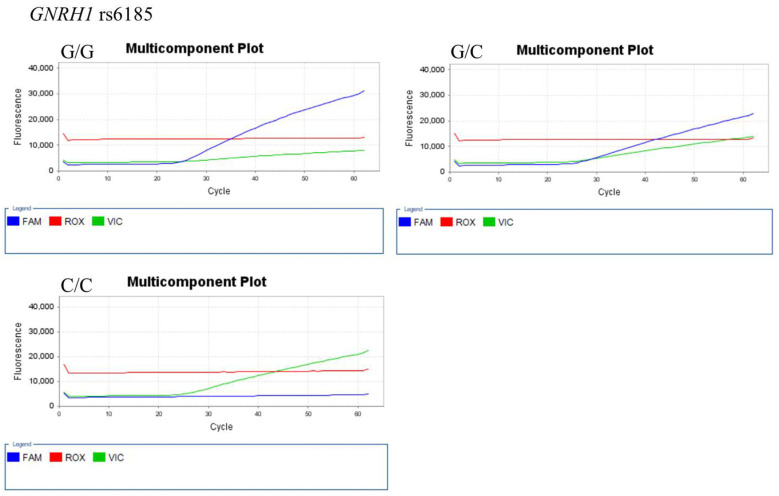
Representative TaqMan assay for GNRH1 (rs6185) genotyping. The FAM (blue) and VIC (green) fluorescence probes detect G and C alleles, respectively. The ROX (red) fluorescence probes is used for calibration.

**Figure 4 ijerph-18-07006-f004:**
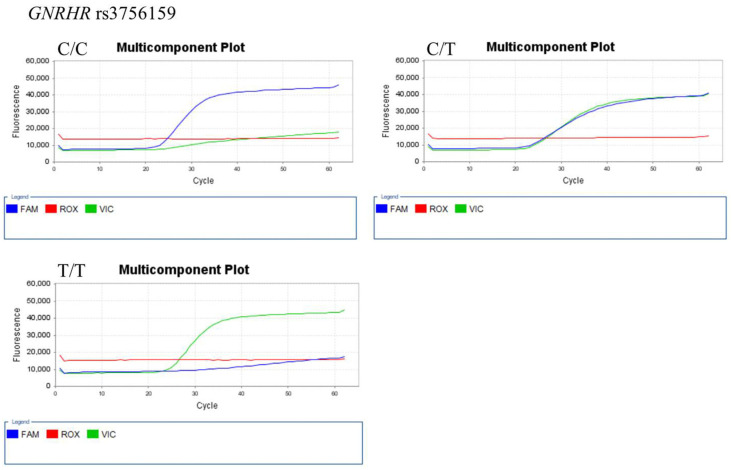
Representative TaqMan assay for *GnRHR* (rs3756159) genotyping. The FAM (blue) and VIC (green) fluorescence probes detect C and T alleles, respectively. The ROX (red) fluorescence probes is used for calibration.

**Figure 5 ijerph-18-07006-f005:**
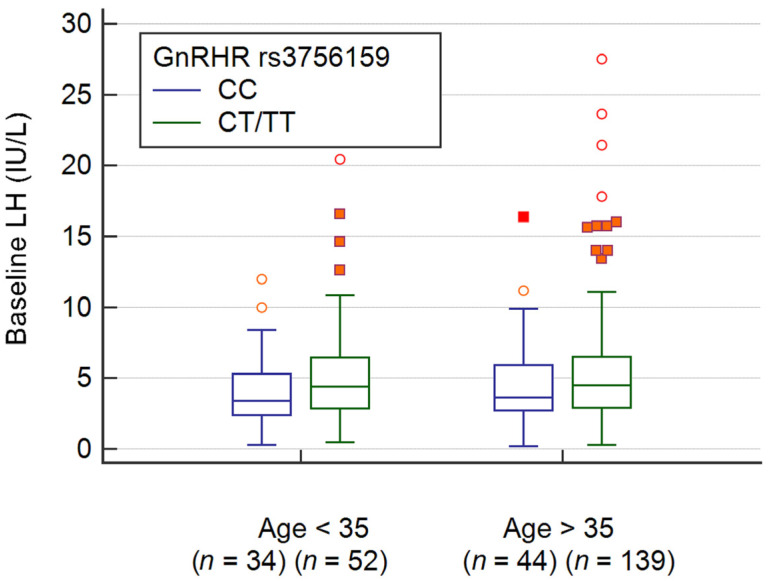
The baseline LH levels for patients with various *GnRHR* rs3756159 genotypes. There was no significant difference of these LH levels between CC vs. CT/TT genotypes in young (POSEIDON group 3, *p* = 0.095) or advanced age (POSEIDON group 4, *p* = 0.152) patients by Mann–Whitney U test.

**Table 1 ijerph-18-07006-t001:** The context sequence of four primers to detect *AMH*, *FSHR*, *GnRH*, and *GnRHR* SNPs in the study.

Variable	Assay ID	Context Sequence
*AMH* 146 T > G (rs10407022)	C__25599842_10	GAAGACTTGGACTGGCCTCCAGGCA[G/T] CCCACAAGAGCCTCTGTGCCTGGTG
*FSHR* A2039G (rs6166)	C___2676874_10	AGGGACAAGTATGTAAGTGGAACCA[C/T] TGGTGACTCTGGGAGCTGAAGAGCA
*GnRH*-1 (rs6185)	C___1529427_1_	CTGGCTGGAGCAGCCTTCCACGCAC[C/G] AAGTCAGTAGAATAAGGCCAGCTAG
*GnRHR*-1 (rs3756159)	C__27477550_10	AACATGAAAGGTATAAAGCCCTCAA[A/G] TGCAGGGTGTGGCTATGAAAGTCGG

**Table 2 ijerph-18-07006-t002:** The frequencies of *AMH* 146T > G (rs10407022) SNP among women with serum *AMH* < 1.2 ng/mL in varied age groups.

*AMH* 146 T > G (rs10407022)	Age < 35 Years (*n* = 86)		Age ≥ 35 Years (*n* = 183)		*p* Value ^1^
Genotype	*n*	%	*n*	%	
TT	37/86	43.0	62/183	33.9	Reference
TG	37/86	43.0	90/183	49.2	0.1913
GG	12/86	14.0	31/183	16.9	0.2773
Recessive					
TT	37/86	43.0	62/183	33.9	Reference
TG/GG	49/86	57.0	121/183	66.1	0.1478
Allele					
T	111/172	64.5	214/366	58.5	Reference
G	61/172	35.5	152/366	41.5	0.1802

^1^ by Chi-square test.

**Table 3 ijerph-18-07006-t003:** The frequencies of *FSHR* A2039G (rs6166) SNP among women with serum *AMH* < 1.2 ng/mL in varied age groups.

*FSHR* A2039G (rs6166)	Age < 35 Years (*n* = 86)		Age ≥ 35 Years (*n* = 183)		*p* Value ^1^
Genotype	*n*	%	*n*	%	
AA	30/86	34.9	73/183	39.9	Reference
AG	48/86	55.8	91/183	49.7	0.3746
GG	8/86	9.3	19/183	10.4	0.9593
Recessive					
AA	30/86	34.9	73/183	39.9	Reference
AG/GG	56/86	65.1	110/183	60.1	0.4316
Allele					
A	108/172	62.8	237/366	64.8	Reference
G	64/172	37.2	129/366	35.2	0.6582

^1^ by Chi-square test.

**Table 4 ijerph-18-07006-t004:** The frequencies of *GnRH*-1 (rs6185) SNP among women with serum *AMH* < 1.2 ng/mL in varied age groups.

*GnRH*-1 (rs6185)	Age < 35 Years (*n* = 86)		Age ≥ 35 Years (*n* = 183)		*p* Value ^1^
Genotype	*n*	%	*n*	%	
GG	25/86	29.1	54/183	29.5	Reference
GC	43/86	50.0	93/183	50.8	0.9966
CC	18/86	20.9	36/183	19.7	0.8387
Recessive					
GG	25/86	29.1	54/183	29.5	Reference
GC/CC	61/86	70.9	129/183	70.5	0.9414
Allele					
G	93/172	54.1	201/366	54.9	Reference
C	79/172	45.9	165/366	45.1	0.8539

^1^ by Chi-square test.

**Table 5 ijerph-18-07006-t005:** The frequencies of *GnRHR*-1 (rs3756159) SNP among women with serum *AMH* < 1.2 ng/mL in varied age groups.

*GnRHR*-1 (rs3756159)	Age < 35 Years (*n* = 86)		Age ≥ 35 Years (*n* = 183)		*p* Value ^1^
Genotype	*n*	%	*n*	%	
CC	34/86	39.5	44/183	24.0	Reference
CT	32/86	37.2	94/183	51.4	0.0071
TT	20/86	23.3	45/183	24.6	0.1166
Recessive					
CC	34/86	39.5	34/183	24.0	Reference
CT/TT	52/86	60.5	139/183	76.0	0.0091
Allele					
C	100/172	58.1	182/366	49.7	Reference
T	72/172	41.9	184/366	50.3	0.0732

^1^ by Chi-square test.

**Table 6 ijerph-18-07006-t006:** The demographic and ovarian stimulation characteristics of infertile woman in *GnRHR* (rs3756159) SNP with CC vs. CT/TT genotype. The data are presented with median (25% to 75%).

Characteristics of Patients	*GnRHR* rs3756159 CC *n* = 78	*GnRHR* rs3756159 CT/TT *n* = 191	*p* Value
Woman age (years)	36.0 (34.0 to 41.0)	38.0 (35.0 to 41.0)	0.0513
Duration of infertility (years)	3.0 (2.0 to 4.0)	3.0 (1.5 to 5.0)	0.5970
Baseline *AMH* (ng/mL)	0.60 (0.43 to 0.94)	0.60 (0.28 to 0.90)	0.2619
Baseline FSH (IU/L)	7.60 (4.76 to 9.50)	7.40 (5.35 to 10.38)	0.6122
Baseline LH (IU/L)	3.60 (2.45 to 5.40)	4.40 (2.91 to 6.48)	0.0308
Baseline E2 (ng/mL)	37.0 (25.0 to 59.0)	37.0 (22.0 to 66.5)	0.8942
E2 on Day of trigger (ng/mL)	775.5 (485.0 to 1267.0)	823.0 (524.8 to 1208.0)	0.4533
P4 on Day of trigger (pg/mL)	0.66 (0.37 to 0.90)	0.66 (0.46 to 1.02)	0.3235
Day of stimulation (days)	14 (13 to 15)	14 (13 to 15)	0.8692
Number of retrieved oocytes	4 (3 to 7)	4 (3 to 6)	0.4028
Number of mature oocytes	3 (2 to 6)	3 (2 to 5)	0.2512
Number of Day3 embryos	3 (2 to 5)	3 (2 to 5)	0.3711
Day3 good embryo rate (%)	70.8 (50.0 to 100.0)	66.7 (50.0 to 100.0)	0.4837

## Data Availability

The data presented in this study are available in Appendix A.

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
