# Peer review of "Association between GnRH Receptor Polymorphisms and Luteinizing Hormone Levels for Low Ovarian Reserve Infertile Women"

_ijerph, 2021, doi:10.3390/ijerph18137006_

Round 1

Reviewer 1 Report

Comments to the manuscript “Association between GnRH receptor polymorphisms and luteinizing hormone levels for low ovarian reserve infertile women”.

The paper is original, well written and the conclusions are supported by the results. I only have the following minor comments:

-Seems that Table I is incomplete: the percentages are missing.

-In line 159 it is indicated that “only the baseline LH levels [3.45 (2.40 to 5.30) vs. 4.40 (2.60 to 6.48)] are different between CC and CT/TT genotype of GnRHR SNP (rs3756159).” Please note that the baseline LH levels are different from those listed in the Table: [3.60 (2.45 to 5.40) vs 4.40 (2.91 to 6.48)]. Please correct the abstract (line 28) too: change 2.405 to 2.45.

Author Response

Thank you very much for the valuable comments. My responses are in an italic style that is attached in the following paragraphs.

The paper is original, well written, and the conclusions are supported by the results. I only have the following minor comments:

-Seems that Table I is incomplete: the percentages are missing.

Response: Thank you for the precious comment. I have modified Table 1 and add the percentages (Table 2, 3,4,5 in the revised manuscript).

-In line 159 it is indicated that “only the baseline LH levels [3.45 (2.40 to 5.30) vs. 4.40 (2.60 to 6.48)] are different between CC and CT/TT genotype of GnRHR SNP (rs3756159).” Please note that the baseline LH levels are different from those listed in the Table: [3.60 (2.45 to 5.40) vs 4.40 (2.91 to 6.48)]. Please correct the abstract (line 28) too: change 2.405 to 2.45.

 Response: Thank you for the specific comment. I corrected line 159 “only the baseline LH levels [3.60 (2.45 to 5.40) vs 4.40 (2.91 to 6.48] are different between CC and CT/TT genotype of GnRHR SNP (rs3756159).”. The typing error in the abstract (line 28) was also corrected.

Reviewer 2 Report

The hormone detection, the primers for the SNP, and how to identify the genotype need to be involved in Materials and Methods part, and the representative figure of each genotype need to be shown in the result part.

The formation of table 1 need to improved. It is not completed, and the authors need to include the percent in it and separate each SNP in different tables and mark the significant difference for each genotype.

Materials and Methods:

  1. List the hormone detection method
  2. List the primers for the SNP loci
  3. The genotyping method

Result:

  1. Show the representative figure of each genotype

    2 . Table 1 is not completed, and the authors need to include the percent

  1. Separate each SNP in different tables and mark the significant difference for each genotype.

Author Response

Thank you very much for your important comments. My responses to the comments are listed at an italic style in the following paragraphs.

The hormone detection, the primers for the SNP, and how to identify the genotype need to be involved in Materials and Methods part, and the representative figure of each genotype need to be shown in the result part.

The formation of table 1 need to improved. It is not completed, and the authors need to include the percent in it and separate each SNP in different tables and mark the significant difference for each genotype.

Materials and Methods:

  1. List the hormone detection method
  2. List the primers for the SNP loci
  3. The genotyping method

Response: Thank you for the valuable comments. I listed the hormone detection methods in an extra paragraph within the section of materials and methods.

The primers for the SNP loci are listed in Tabel 1.

The genotyping methods are real-time PCR based that performed following the instructions of commercial kits.

Result:

  1. Show the representative figure of each genotype
  2. Table 1 is not completed, and the authors need to include the percent
  3. Separate each SNP in different tables and mark the significant difference for each genotype.

Response: Thank you for the precious comments.

We demonstrated the representative figure of each genotype in Figures 1, 2, 3, and 4.

The original Table 1 is divided into Table 2, 3, 4, and 5 in the revised manuscript and the percentage of genotypes and alleles are included in the separated tables.

Round 2

Reviewer 2 Report

The figure and  table is not completed. 

Please explain the FAM ROX VIC in the figure legends of Fig.1 to Fig.4

Please complete the percentage of each genotype in Table 2 to Table 5, and the P value.

Author Response

Thank you for the valuable comments. Please see the attached file for our responses to the comments.

This manuscript is a resubmission of an earlier submission. The following is a list of the peer review reports and author responses from that submission.